# Walking and Playing with the Dog to Improve the Physical Activity Level of Adolescents: A Scoping Review

**DOI:** 10.3390/healthcare12060631

**Published:** 2024-03-11

**Authors:** Simona Pajaujiene, Luca Petrigna

**Affiliations:** 1Department of Coaching Science, Lithuanian Sports University, 44221 Kaunas, Lithuania; simona.pajaujiene@lsu.lt; 2Department of Biomedical and Biotechnological Sciences, Human, Histology and Movement Science Section, University of Catania, Via S. Sofia n°87, 95123 Catania, Italy

**Keywords:** pet, wellness, health, animal, training

## Abstract

Background: In recent years, new technologies such as the introduction of the smartphone and the tablet in everyday life and habits have often made adolescents sedentary. It is becoming a serious problem in society. It is important to propose, as soon as possible, proper and feasible programs to modify this trend. A solution should be to increase physical activity, reducing sedentary behaviors. Adopting dogs and walking and playing with them could be a solution, and the population should be sensitized about this aspect. Consequently, the objective of this scoping revision of the literature is to analyze the association between physical activity in adolescents and the presence of a dog in the family. Methods: Three electronic databases were screened until 21 February 2024. The detected articles were screened against the eligibility criteria. The results were narratively discussed. Results: After the screening process, a total of four studies were included. The studies presented heterogeneity in the physical activity assessment. This made it hard to synthesize the results. Indicatively, there is a positive association between physical activity and dog ownership. Conclusions: walking and playing with the dog increase the physical activity level of adolescents. Therefore, a sensibilization campaign should highlight the importance of having dogs in families, especially in adolescence. It is fundamental to daily walk and play with them.

## 1. Introduction

The world health data is alarming: in 2030, 38% of adults will be overweight [1] and 20% obese [1]; the number of people with type 2 diabetes is expected to rise to 552 million [2]; and 23 million people are expected to die from cardiovascular disease [3]. It is well known that obesity is associated with an increased risk of mortality and morbidity for diabetes mellitus, hypertension, cardiovascular disease and stroke, osteoarthritis, and cancer [1]. Obesity is associated with sedentary behaviors such as the time spent in front of a screen [4,5]. Screen viewing is associated with higher consumption of energy foods [6], further increasing the health risks. Generally, physical inactivity causes more than 3 million deaths per year [7,8].

It is known that about one-third of people do not meet the recommended standards for cardiorespiratory fitness [9]. The problem starts in childhood [10]. It continues in adolescence; indeed, the majority of them present an insufficient physical activity level [11]. Worldwide, about 80% of people aged 13–15 years do not reach the recommended daily 60 min of physical activity [12]. This percentage decreases to about 31% in adults [12]. From the above information, it seems that adolescence is a critical age that requires a proper evaluation and a feasible intervention. The evaluation of physical activity levels can be performed with self-report tools such as questionnaires. They can be proposed as short-term recall questionnaires (with several specific questions in a short period) such as the International Physical Activity Questionnaire [13], global questionnaires (with few generic questions) such as the Stanford Brief Activity Survey [14], qualitative history recall questionnaires (with several questions on the subject of last year/lifetime physical activities) such as the Minnesota Leisure Time Physical Activity Questionnaire [15], or the Physical Activity logs (about information about short activities completed during the day) such as the Bouchard Physical Activity Log [16]. Another serf-report tool to assess physical activity levels is the diary of short (from one day) and long (several weeks) duration [17]. To evaluate physical activity objectively, wearable monitors can be adopted, such as pedometers (to identify the number of steps), accelerometers (to collect data about movement and sedentary behaviors), and heart rate monitors [17]. The pedometers are easy to adopt and cheap instruments that present validity, especially in large-scale epidemiological studies, in children and adolescents [18]. The technology of accelerometers is more complex; they collect information about acceleration on the planes in gravitational units, but they are not always very accurate [19]. Heart rate monitors can also be used to estimate energy expenditure. While they are precise in detecting the heart rate, they present some limits in the energy expenditure monitoring [20]. The physical activity assessment tools are various and different, with different accuracy and utility; they should be adopted according to the population characteristics [17]. The topic “evaluation” is deeply studied, and it is evolving faster. On the other side, intervention still requires attention.

The main objective of Governments should be to intervene as soon as possible on the fitness and cardio-metabolic health, adiposity, and behavioral conduct of the citizens. They should find feasible and valid solutions to limit sedentary time, especially during childhood [21]. In theory, children and youth should do at least 60 min per day of moderate physical activity with vigorous intensity activities incorporated into the training, mostly aerobic but also with muscle strength training and stretching activities [21,22]. In this way, it is possible to obtain important benefits in cardiorespiratory and musculoskeletal fitness, dyslipidemia, glucose and insulin resistance, cognitive outcomes (such as academic performance and executive function), and mental health [21,23]. Sedentary behaviors adopted during the day increase the risk of obesity [24], therefore programs are required to reduce these unhealthy behaviors involving schools and families [25,26,27]. The interventions to reduce a sedentary lifestyle are different and various [28,29,30], but the trend is not changing. It was suggested that interventions should be based on human behaviors changing [31], and walking a dog could be a possible solution. 

Dogs and human health are often associated. Dogs could be trained as physical service dogs, diabetes-alert dogs, epilepsy-alert dogs, and hearing dogs [32]. The four paw companions are trained to meet the person’s needs, increase their independence, and decrease the need for health and social care [32]. Studies also demonstrated that having a pet at home improves perceived health [33], and the interaction with the dog, such as the tactile sensation, improves the owner’s well-being [34]. They indirectly improve cardiovascular system health with a decrease in the prevalence of systemic hypertension [35]. People with pets, generally present higher levels of overall life satisfaction, especially if they have dogs and cats [36]. Having a dog in the family improves the social (practical and emotional) relationships with the neighbors whether incidental or appositely created [37]. Pet exposure during development was negatively associated with hypertension and blood pressure [38]. People who generally walk with a dog are generally less obese [39], but some studies detected contradictory findings [40,41]. This aspect needs to be further explored, also at different ages, such as adolescents. 

Adopting a dog in the family has an important role in the level of physical activity, and it is also good for health [42]. Their presence in a family is associated with more time spent walking, especially for those who live close to a green area [42]. People who decide to adopt a dog generally present more frequent leisure-time physical activity [43]. Out of the people who take walks with their dogs, 80.2% of them spend at least 10 min marching with them, while 42.3% walk for 30 min daily [44]. A study found that they spent 22 min a day walking [45]. Similarly, studies detected that dog owners walked about 18 min per week more than people with no dogs [46,47] or more than 100 min per week [48]. People with a dog reach, easily and with a higher percentage, the recommended 150 min per week of moderate to vigorous physical activity [39,49,50,51,52]. This is confirmed by other studies in which people with a dog spent from 160 [53] to 300 min per week in mild-moderate physical activities (non-dog owners spent only 160 min per week) [54]. In a study, people with dogs did from 14 to 19 min per week of moderate- and vigorous-intensity physical activity more than non-dog owners [55]. Similar findings were also detected in children [56,57,58], older adults [59], and pregnant women [40]. Considering all the previously introduced aspects, the objective of the present study was to review the literature to analyze the association between physical activity levels in adolescents and the presence of a dog in the family.

## 2. Materials and Methods

The scoping review partially adopted the Preferred Reporting Items for Systematic Reviews and Meta-Analyses Extension for Scoping Reviews (PRISMA-ScR) checklist and explanation [60]. The protocol was not previously registered on PROSPERO, but it was written down before the beginning of the work.

### 2.1. Eligibility Criteria

Population, intervention, comparison, outcomes, and study design (PICO-S) points were followed as suggested by the PRISMA-ScR checklist and explanation [60]. Studies were included if the sample was composed of healthy (physically and mentally) adolescents. The reference years were 12–18 years old, as the literature suggests [61] (population). Regarding the intervention, the studies had to specify the presence of the dog in the everyday life of the sample studied. For comparison, the data of a group composed of people with a dog had to be compared with the data of people without dogs. The outcomes considered were about physical activity level. All typologies of study design (i.e., randomized controlled trials, clinical trials, and longitudinal studies) were included. The language adopted in the included studies had to be English; studies written in other languages were not considered. Regarding the country of origin, no limitation was adopted. Studies were included only if papers were original, peer-reviewed, and published in international journals. 

### 2.2. Information Sources

Articles were collected from PubMed, Web of Science, and Scopus electronic databases. The screening of the articles was performed until 21 February 2024. To search the articles, different keywords were adopted and matched. The keywords adopted to create the search string were:Keywords 1: dog, pet, dog owner, puppy;Keywords 2: physical activity, exercise, sport;Keywords 3: adolescent, teenage, young, young adult.


Through the Boolean indicator “AND” or “OR” the following string was generated:

(dog OR pet OR “dog owner” OR puppy) AND (physical activity OR exercise OR sport) AND (adolescent OR teenage OR young OR “young adult”).

This string was used in the three electronic databases searched.

### 2.3. Data Selection and Management

After the search, all articles detected were collected. Duplicates were deleted through the program EndNote X8 (EndNote version X8; Thompson Reuters, New York, NY, USA). Some of the duplicates were removed manually in a second step. A manual section performed by the two investigators by title, abstract, and full-text was performed to exclude the articles according to the eligibility criteria. The steps of this process are presented in the PRISMA flow diagram, as suggested by the PRISMA-ScR checklist and explanation [60].

### 2.4. Data Collection and Synthesis

A Microsoft Excel spreadsheet (Microsoft Corp; Redmond, WA, USA) was created to extract the following information: first author and year of publication of the study; sample size and participant gender and age (range, or mean and standard deviation); tests adopted to detect physical activity level, and the study’s main results. The data were summarized using tables, and the findings were analyzed through a narrative synthesis.

## 3. Results

A total of 2132 studies were detected in the three databases searched after the search. A number of 809 studies were on PubMed, 409 on Scopus, and 914 on Web of Science. After the duplicate’s removal, a total of 1664 articles remained for the eligibility criteria selection. After the screening against the inclusion and exclusion criteria, a total of 4 articles were suitable for the study, and consequently, they were included. Detailed information on the screening process is in the flow diagram in Figure 1, as suggested by PRISMA [62].

### 3.1. Participants Characteristics of the Included Studies

A total of 3747 participants were included in the review. It has not been possible to divide the participants according to gender due to the lack of this information in some of the studies. The age ranged between 12 and 17 years. It has not been possible to provide a mean age due to the lack of this information in some of the studies. Poor information was provided about the participants’ daily routing, limiting the possibility of analyzing the findings based on the sample characteristics. Poor information was also provided about the physical and sporting background, creating confusion in the interpretation of the results. Detailed information was detected about the study participants’ characteristics (number of participants in each study and age in terms of mean or age range), physical activity evaluation methods, and main findings in Table 1. 

### 3.2. Studies Characteristics of the Included Studies

The study by Martin and colleagues [63] was on a sample of Australian adolescents recruited in local schools. In their study, Westgarth and colleagues [64] recruited people from the United Kingdom. It is a longitudinal study. The study of Sirard and colleagues [65] is a cohort study performed in the United States of America. Engelberg and colleagues [66] also collected the data in the United States of America, but it was an observational study, and they recruited the participants by email or telephone.

Physical activity was measured with a questionnaire in a study [63]. In particular, the authors Martin and colleagues [63] adopted the modified version of the Children’s Leisure Activities Study Survey questionnaire [67]. It is a self-reported questionnaire on the participation in 40 different activities in the last 7 days, and between them there are also questions about the routine with the dogs. 

Three studies adopted a questionnaire or a survey to obtain information about the sample, the presence of a dog, and an accelerometer such as the ActiGraph (Pensacola, F) [64,65,66]. The ActiGraph is an activity monitor adopted in physical activity research. In two studies [64,65] the accelerometer was placed on the right hip for 7 days. In the study of Engelberg and colleagues [66], participants wore the accelerometer for 1 week with a belt positioned on their left iliac crest. 

In a study [64], the data recorded were the daily average counts per minute of overall physical activity and the daily average in minutes spent in moderate-to-vigorous physical activity [64]. The data were analyzed statistically. The authors of another study [65] calculated the summary of physical activity variables and the total movement. In the last study [66], the authors collected data about the daily time spent in moderate-to-vigorous physical activity. They analyzed the data with statistical analysis. From the protocols adopted, it was hard to provide accurate and specific information. The problem was due to the differences in the surveys proposed, but especially because of differences in the data collection process and in the data adopted by the authors.

### 3.3. Main Findings of the Included Studies

The conclusions of the studies were similar. The study by Martin and colleagues [63] found that young people, due to the active play with the dog, increase their daily activities by about 1 h a week. According to the study [63], about 26% of the adolescents reported dog walking at least once a week. The percentage reached 44% of adolescents who also reported pet play. Martin and colleagues [63] reported that there was a median of 60 min spent in dog walking and pet play in the last week. According to Westgarth and colleagues [64], about 40% of adolescents tend to walk with their dogs from 2 to 6 times a week. In this study [64], the main findings are only subjective; indeed, they are not correlated to objectively measured physical activity. In the study of Sirard and colleagues [65], a positive association (*p* < 0.05) exists between families with dogs and objectively measured physical activity. But no link exists with sedentary behavior. According to this study, mean daily accelerometer counts per minute were significantly associated with dog ownership. Opposite results were obtained in the study of Engelberg and colleagues in 2016 [66]. According to the authors [66], adolescents with dogs had about 5 more minutes of moderate to vigorous physical activity per day if compared to those who did not walk with the dogs [66]. 

## 4. Discussion

The revision of the current literature detected a relationship between physical activity level and the presence of a dog in the lives of adolescents. The scoping review also detected few and heterogeneous studies about this topic, highlighting the necessity for more research to support the role of a dog in the lives of young people. Adolescents who live with a dog or decide to adopt a dog are generally more physically active if compared with peers who do not have a dog. The studies included stated this after a statistical analysis of their data. The findings of this study highlight how important the presence of a dog is in adolescents’ lives. The dog should be part of the family from a young age, but especially in those years when there is a transition period, such as during adolescence. 

According to the World Health Organization, the term health is considered “complete physical, mental and social well-being and not merely the absence of disease or infirmity” [68]. In this regard, a dog could be considered a personal trainer who motivates it to be physically active and provides psychological help and social support. The results suggest that having a dog in the family increases the number of active days, the metabolic equivalent per minute per week, and calorie expenditure [65]. The study by Engelberg and colleagues [66] quantified that adolescents with dogs had about 5 min more of moderate-to-vigorous physical activity per day. If these data also consider the active play of young people, it could even reach 1 h more of weekly activities [63]. Only one study had contradictory results [64]. It is probably because most of the participants (94–87%) reported never walking with their dogs. The heterogeneity in the results made it difficult to deeply discuss the topic. The methods to assess physical activity were subjective, such as questionnaires and surveys, and objective, such as accelerometers. While the instruments adopted were the same, there were differences in the data extracted, making it impossible to compare the results.

According to the literature, having a dog at home has a positive effect on the physical level of the person, as well as in adolescents. The results of the present study are in line with the literature on other populations or that evaluated other aspects [69,70]. A study evaluated the visitors of a park, and they noted that during good weather conditions, there were mixed (with and without a dog) people that went to the park [71]. On these occasions, most of the visitors were in the park to train themselves [71]. Differently, when the weather conditions were not good, most of the people who visited the park were visitors with their dogs [71]. The message of this interesting study is that people with dogs were attending the park anyway, despite the weather conditions [71]. This study is in line with our findings; indeed, adolescents with dogs were more active than people without dogs, despite their daily routine. The findings of the study by Temple and colleagues [71] indirectly suggest a better general health status for people with dogs. Indeed, the park has an important role from a physical, social, and psychological perspective [72,73,74,75]. The park is positively correlated with physical activity levels [73,74], and it decreases the incidence of physical and mental illnesses [76]. Another important aspect among female dog walkers was that they felt safer walking with their dogs in their neighborhood [48]. Both aspects are important when the objective is to associate the presence of a dog with the physical, psychological, and social health of the person. 

According to the literature, the five aspects of health promotion programs are physical and mental health, everyday functioning in social and role activities, and general perceptions of well-being [77]. A dog can satisfy all these aspects. Related to physical well-being, the findings of the present study support their role also in adolescents. In the period of life investigated, the dog can become the everyday routine that guides the population in this life stage characterized by instability [78]. The first aspect to consider is the positive effect from a physical point of view. The necessity of walking with the dog could be a means to reach the 60 min of aerobic daily activities suggested by the American College of Sports Medicine [79]. The benefits also come from a psychological and social point of view. Indeed, in adolescents, having a family pet or sharing pets with their parents may provide a positive influence on loneliness levels, even if this topic is complex and should be further studied [80]. In rural adolescents, pet owners reported significantly lower loneliness scores than non-pet owners [81]. So, beyond keeping people physically active, animals provide social support and companionship to all ages [82], having a kind of social role in the life of the person. Interestingly, a study detected that adolescents are more satisfied with the relationship with their pets than with their brothers, while girls have better companionship with their pets than boys [83]. This further supports the fundamental role of a dog in the life of an adolescent. Unfortunately, the literature on the mental health benefits of having a pet at home is still incomplete. The studies are often heterogeneous and hard to compare [84,85].

The World Health Organization [68] says governments should promote and protect all people’s health; the policies of governments to reduce sedentary behaviors and improve physical activity are usually only somewhat effective [86]. This suggests the necessity of alternative and feasible programs. An intervention in youth is important for the immediate effects and for the modification of behaviors that will continue into adulthood [87]. Children and young adults are subject to risky behaviors such as physical inactivity, smoking, drinking, or poor nutrition [88], making it important to make proper and timely interventions. Considering that dog attachment is stronger among younger adolescents [89], they should obtain a dog as soon as possible, also because the practice of regular physical activity in childhood is an aspect of a healthy future [90]. Governments could raise awareness about the importance of having a dog in the house. They could also promote walking and being physically active with them. The findings of the present study could help support the possible programs proposed to increase physical activity levels.

An aspect to consider is that a dog is not a toy or a hobby; sometimes it could be hard to balance dog ownership with everyday life aspects such as education or work [78]. There are some people with dogs who do not take walks with them [50], ignoring the necessities of their pets and limiting the benefits of an active lifestyle. Consequently, dogs are not a health solution for everyone but only for those who can take care of them and themselves.

The studies included in this review are few; this aspect limits, most importantly, the findings of the present manuscript. On the other hand, the fewer studies on this topic also show the need for future studies, better if longitudinal, that evaluate the positive and, eventually, negative aspects of having a dog in the adolescent’s life. Due to the limited number of studies, it was not possible to stratify the population into smaller groups based on their development stage, gender, physical or sporting activity level, or socio-economic status. Indeed, adolescence is a complex time, and studies should better consider their developmental stage and not only their chronological age. Furthermore, the physical and socio-economic status could be influencing factors in the life of the adolescent. Future studies could better investigate the above aspect. There is also heterogeneity in the methodology (data collection process and data extrapolation) adopted, making it hard to interpret the results. Also, this aspect highlights the necessity of future studies on this topic that follow a standard operating procedure [91] to have the possibility to analyze more consistent and well-structured works. Another limitation is related to the selection process, which was performed by only two authors. Fortunately, both investigators agreed on which articles to include or exclude without having to ask a third researcher. One more limitation is related to the inclusion of only English-written manuscripts; possible articles written in other languages were excluded, limiting the findings of the study.

Most of the studies are self-reported, which may not accurately reflect dog-owner behaviors, as is highlighted in other studies [71]. We can only provide some feedback for future studies and to sensibilize the population about the importance of having a dog in our lives. 

## 5. Conclusions

The presence of a dog in the life of an adolescent has positive outcomes in terms of physical activity level. Walking and playing with the dog increase the weekly physical activity level, reaching the health recommendation. Governments should adopt the present findings to support their programs to fight the decrease in physical activity levels since young ages. This study suggests that adopting a dog during this complex period of life helps the person be more active. The discussion also highlighted the psychological and social benefits of a dog in a person’s life. The findings also suggest the need for future studies on this topic with bigger samples and measuring physical activity levels objectively with standardized measurements. Future studies could also investigate the role of dogs in other populations, such as older adults or people with disabilities. An interesting factor that could be better evaluated could also be the role of a dog in childhood and adult life.

## Figures and Tables

**Figure 1 healthcare-12-00631-f001:**
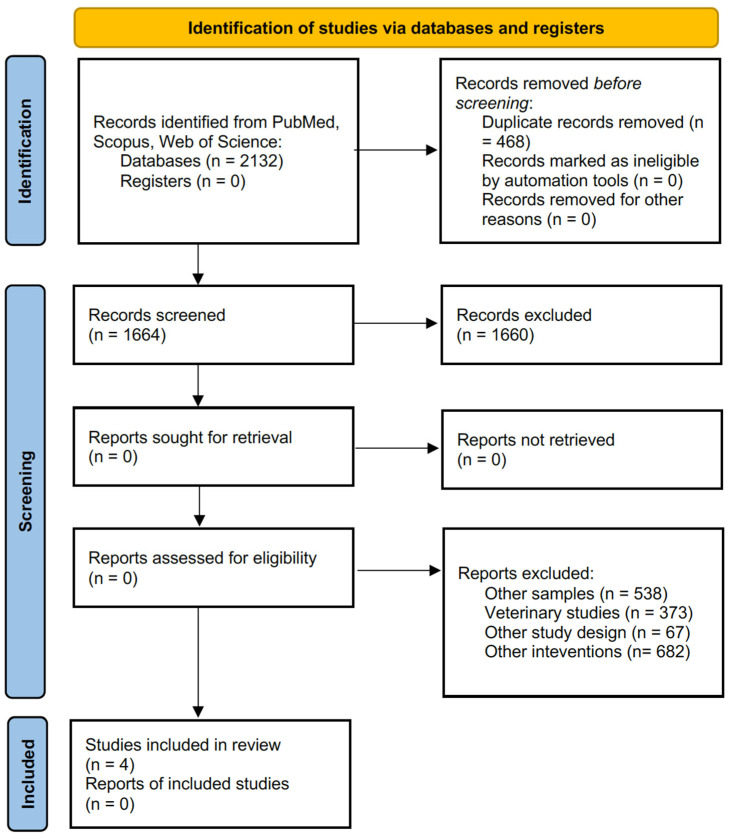
PRISMA 2020 flow diagram.

**Table 1 healthcare-12-00631-t001:** Description of studies.

First Author, Year	Number	Age Years	Evaluation	Main Findings
Martin, Wood et al., 2015 [63]	657	14.0 (1.3)	Questionnaire	The median of 60 min of dog walking in the past 7 days
Westgarth, Ness et al., 2017 [64]	1547	15–16	ActiGraph accelerometer	Dog walking increases during adolescence. Walk 2–6 times/week (39–46%) or never (27–37%). 7–8% reported to walk every day
Sirard, Patnode et al., 2011 [65]	618	14.6 (1.8)	ActiGraph accelerometer	Mean daily minutes Moderate-Vigorous Physical Activity was significantly greater
Engelberg, Carlson et al., 2016 [66]	925	12–17	Survey data andaccelerometer	4–5 min or more of Moderate-Vigorous Physical Activity

## Data Availability

All data used are within the manuscript.

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
