# Peer review of "Walking and Playing with the Dog to Improve the Physical Activity Level of Adolescents: A Scoping Review"

_healthcare, 2024, doi:10.3390/healthcare12060631_

Round 1
Reviewer 1 Report
Comments and Suggestions for Authors
Thank you for your manuscript. Your topic is interesting. However, you need quite work to improve your manuscript. Please see my comments below… You need to revise your text… Moderate editing of English language required…
TITLE
The current title leads the reader to believe that this is a scoping review on interventions, but in reality it is not... I suggest changing the title: “Dog-Ownership to Improve The Physical Level of Adolescents: A Scoping Review”.
ABSTRACT
P1L10. “Background: Sedentary behavior is becoming a serious problem for the society increasing the risk of disease and death. It is important to propose as soon as possible, proper and feasible intervention program to modify this negative trend.” – I believe the background should be focused on your aim, i.e., physical activity in adolescents…
P1L14. “Consequently, the objective of the present study, through a revision of the literature, is to analyze the association between physical activity and dog walking in adolescent and young adults. Methods: This scoping review aimed to investigate the association between dog-owing and physical activity levels in adolescents and young adults.” – Why do you need two sentences to describe the aim of the study?
INTRODUCTION
P1L29. “…in 2030, in 29 the world, major depression will be the first cause of burden of disease [3]…” – The importance of this sentence for the article is not clear...
P2L44. “The literature [14] suggests that between 15 and 18 years of age there is a general decrease in physical activity level; it continues until about 29 years of age.” – This study was conducted in the USA. Can we generalize to the rest of the world?
4th paragraph – The importance of this paragraph in relation to the objective of the study is not understood – to study the association between dog ownership and physical activity…
Indeed, this Introduction section is a loosely organized mix of benefits associated with having a dog at home, many of them with no obvious connection to the objective of the study, i.e., to study the association between dog ownership and physical activity…
It is not clear in Introduction section why adolescents and young adults were chosen as the target population. They are populations with different characteristics. In fact, exactly this is discussed in the Discussion section. Moreover, the last paragraph describe benefits for children who have a dog. Children should be considered other population…
P3L114. “…recognizing that the critical age is between late adolescence and during young adulthood…” – Reference… Why is this the critical age?
I believe the aim of the study is not clearly described… It should be more concise and not give rise to double interpretations...
METHODS
The ages that defined being or not being a adolescent must be described...
P3L132. “…physical activity had to be evaluated subjectively through questionnaires and surveys; or objectively through accelerometers.” – Why these measures? There is nothing in the Introduction section nor in the Materials and Methods section on this issue...
P3L134. “The language adopted in the included studies had to be English, studies written in other languages were not considered.” – This must be considered a limitation of the study…
P4L148. “(dog OR pet OR "dog owner" OR puppy) and (physical activity OR exercise OR sport) and (adolescent OR teenage OR young OR "young adult")” – change to:”(dog OR pet OR "dog owner" OR puppy) AND (physical activity OR exercise OR sport) AND (adolescent OR teenage OR young OR "young adult")”.
How many researchers carried out the studies selection? Possible limitation of this study…
How many researchers carried out the data collection of the selected studies? Possible limitation of this study…
Was no tool used to assess the quality of the included studies?
RESULTS
With only 6 studies selected, I think the table should have more information about them...
P6L200. “According to Westgarth, Ness and colleagues in 2017 [64], adolescents tend to increase dog walking and dog ownership.” - It's not clear what you want to convey in this sentence…
The results must be presented in a more concise and clear way...
DISCUSSION
P6L213. “The revision of the current literature, detected a correlation between physical activity and the presence of a dog in the life of adolescence or young adults.” – I believe that “correlation” is not the better word in this case…
In the Discussion section there is no discussion about the methods used to assess physical activity...
P8L295. “There is also heterogeneity in the population included…” – This heterogeneity is the result of your methodological choices... see previous comments on the chosen target population...
Comments on the Quality of English LanguageModerate editing of English language required…
Author Response
Thank you for your manuscript. Your topic is interesting. However, you need quite work to improve your manuscript. Please see my comments below… You need to revise your text… Moderate editing of English language required…
Thank you very much for the revision and for your time in this manuscript. We really appreciated the Reviewer comments and feedbacks. The manuscript has been importantly edited and the English has been reviewed. We hope the Reviewer appreciated all the work done.
TITLE
The current title leads the reader to believe that this is a scoping review on interventions, but in reality it is not... I suggest changing the title: “Dog-Ownership to Improve The Physical Level of Adolescents: A Scoping Review”.
Reply: We modified the title according the Reviewers suggestion.
ABSTRACT
-P1L10. “Background: Sedentary behavior is becoming a serious problem for the society increasing the risk of disease and death. It is important to propose as soon as possible, proper and feasible intervention program to modify this negative trend.” – I believe the background should be focused on your aim, i.e., physical activity in adolescents…
Reply: Thank you for this comment, we modified the first sentence of the abstract.
P1L14. “Consequently, the objective of the present study, through a revision of the literature, is to analyze the association between physical activity and dog walking in adolescent and young adults. Methods: This scoping review aimed to investigate the association between dog-owing and physical activity levels in adolescents and young adults.” – Why do you need two sentences to describe the aim of the study?
Reply: Thank you for the comment, we deleted the aim in the methods section.
INTRODUCTION
P1L29. “…in 2030, in 29 the world, major depression will be the first cause of burden of disease [3]…” – The importance of this sentence for the article is not clear...
Reply: Thank you for this point, we removed the reference, changed the structure to better highlight the world situation.
P2L44. “The literature [14] suggests that between 15 and 18 years of age there is a general decrease in physical activity level; it continues until about 29 years of age.” – This study was conducted in the USA. Can we generalize to the rest of the world?
Reply: Thank you for this point, we modified the sentence to better highlight the world situation and added references.
-4th paragraph – The importance of this paragraph in relation to the objective of the study is not understood – to study the association between dog ownership and physical activity… Indeed, this Introduction section is a loosely organized mix of benefits associated with having a dog at home, many of them with no obvious connection to the objective of the study, i.e., to study the association between dog ownership and physical activity…
Reply: In this section we wanted to introduce dogs’ benefits in different populations. We were of the idea that a better view of the role of the dogs, could better help to understand the importance of them in our life. We modified this part to keep only the most important and pertinent information, we hope the reviewer appreciate the work done.
-It is not clear in Introduction section why adolescents and young adults were chosen as the target population. They are populations with different characteristics. In fact, exactly this is discussed in the Discussion section. Moreover, the last paragraph describe benefits for children who have a dog. Children should be considered other population…
Thank you, after the reviewer consideration, we focalized the attention only on adolescents. We hope the reviewer appreciate how the manuscript have been changed.
-P3L114. “…recognizing that the critical age is between late adolescence and during young adulthood…” – Reference… Why is this the critical age?
Reply: Thank you for this point, we focalized the attention only in adolescent. As the literature suggest, in this period of life there is a huge number of people that do not reach the physical activity guidelines,
-I believe the aim of the study is not clearly described… It should be more concise and not give rise to double interpretations...
Reply: Thank you for this point, we made our hypothesis more concise and easier to follow.
METHODS
-The ages that defined being or not being a adolescent must be described...
Reply: Thank you. We added it, thank you.
-P3L132. “…physical activity had to be evaluated subjectively through questionnaires and surveys; or objectively through accelerometers.” – Why these measures? There is nothing in the Introduction section nor in the Materials and Methods section on this issue...
Thank you for this comment, we removed this information not previously presented.
-P3L134. “The language adopted in the included studies had to be English, studies written in other languages were not considered.” – This must be considered a limitation of the study…
Reply: Thank you, we added it.
-P4L148. “(dog OR pet OR "dog owner" OR puppy) and (physical activity OR exercise OR sport) and (adolescent OR teenage OR young OR "young adult")” – change to:”(dog OR pet OR "dog owner" OR puppy) AND (physical activity OR exercise OR sport) AND (adolescent OR teenage OR young OR "young adult")”.
Reply: Thank you, we corrected it.
-How many researchers carried out the studies selection? Possible limitation of this study… How many researchers carried out the data collection of the selected studies? Possible limitation of this study…
Reply: Thank you for this point, we specified that the selection process has been performed by the two authors as we did and it has been added in the limit of the study, as suggested, this possible limitation. Related to the data collection, it has been performed by both authors, fortunately, the manuscript included were not a lot simplifying the work.
-Was no tool used to assess the quality of the included studies?
Reply: No, thank you for the question. We decided to structure the review as a scoping review following PRISMA structure for scoping review that doesn’t require the quality assessment. We took this decision to include as much articles as possible.
RESULTS
With only 6 studies selected, I think the table should have more information about them...
Reply: thank you for the suggestion, we added information in the results section.
P6L200. “According to Westgarth, Ness and colleagues in 2017 [64], adolescents tend to increase dog walking and dog ownership.” - It's not clear what you want to convey in this sentence…
Reply: thank you for the comment. We modified the sentence. We hope the reviewer appreciate how it is now the sentence.
The results must be presented in a more concise and clear way...
Reply: thank you for the suggestion. The method section has been improved according the reviewer suggestion.
DISCUSSION
-P6L213. “The revision of the current literature, detected a correlation between physical activity and the presence of a dog in the life of adolescence or young adults.” – I believe that “correlation” is not the better word in this case…
Reply: thank you for the comment. We changed “correlation” with “relation”, we hope the reviewer agree with as that it is the appropriate world.
In the Discussion section there is no discussion about the methods used to assess physical activity...
Reply: thank you for the comment, we added information about the methods adopted and the limits in the assessment.
-P8L295. “There is also heterogeneity in the population included…” – This heterogeneity is the result of your methodological choices... see previous comments on the chosen target population...
Reply: Thank you for this comment, we decided to focalize the attention only in adolescents. In this way we had a more homogeneous population.
Reviewer 2 Report
Comments and Suggestions for Authors
First of all, I congratulate you for the work done. In my opinion, some suggestions can be made:
Abstract
At the end conclusion – what information your study bring.
Introduction
It must be specified whether the data of the study (which was in the United States of America) are also relevant for other geographical regions (countries)
Present more clearly the population you are referring to.
3.3. Main findings of the included studies
This section should be presented more extensively. I couldn't find the results of the study that used the accelerometer.
Conclusions:
At the end of this section, recommendations for future research should be included.
Author Response
First of all, I congratulate you for the work done. In my opinion, some suggestions can be made:
Thank you very much for the revision and for your time in this manuscript. We really appreciated the Reviewer comments and feedbacks. We hope the work done is appreciated by the Reviewer.
Abstract
-At the end conclusion – what information your study bring.
Reply: thank you for the comment, we added a more concise concept.
Introduction
It must be specified whether the data of the study (which was in the United States of America) are also relevant for other geographical regions (countries)
Reply: Thank you for this point, we modified the sentence to better highlight the world situation and added references.
-Present more clearly the population you are referring to.
Reply: Thank you for this comment, we decided to focalize the attention only in adolescents. In this way we had a more homogeneous population. We also added a reference in this regard.
3.3. Main findings of the included studies
This section should be presented more extensively. I couldn't find the results of the study that used the accelerometer.
Reply: thank you for the suggestion. We modified the results section adding more information and data. We hope the authors appreciate the work done.
Conclusions:
At the end of this section, recommendations for future research should be included.
Reply: thank you, we implemented the conclusion with a sentence for future studies.
Reviewer 3 Report
Comments and Suggestions for Authors
Overall, the introduction is well-written based on the key literature. However, it would be beneficial to present the limitations of previous studies to convincingly describe the need and objective of the current study. Additionally, the authors should provide more explanations for why they chose the population between the ages of twelve and thirty. In developmental psychology, these age groups are not typically considered the same. Furthermore, there may be age-related discrepancies that need to be addressed. Please add explanations of these findings in the Discussion section. Additionally, consider the following suggestions to enhance this manuscript prior to publication.
Please clarify the meaning of the sentence in line 134 that reads 'The studies were included despite the study design.' Consider rephrasing for clarity.
In line 154, please add a period after 'in a second step, manually.'
In the discussion section, please clarify whether the differences mentioned in the previous studies are descriptive or statistically significant mean differences.
In line 243, the study by Temple and colleagues [69] has indirect implications. Please clarify the nature of these implications.
In line 246, please rewrite the sentence for clarity.
In line 270, the authors mention several negative findings. Please provide additional explanations of these findings.
In line 275, when designing interventions for youth, it is important to consider their developmental stage. This is one of the reasons why physical activity guidelines for youth are more stringent than for other populations. Please add the implications of this to the literature.
In lines 281-291, this section appears somewhat arbitrary. Please revise and simplify the language.
Author Response
Overall, the introduction is well-written based on the key literature. However, it would be beneficial to present the limitations of previous studies to convincingly describe the need and objective of the current study. Additionally, the authors should provide more explanations for why they chose the population between the ages of twelve and thirty. In developmental psychology, these age groups are not typically considered the same. Furthermore, there may be age-related discrepancies that need to be addressed. Please add explanations of these findings in the Discussion section. Additionally, consider the following suggestions to enhance this manuscript prior to publication.
Thank you very much for the revision and for your time in this manuscript. We really appreciated the Reviewer comments and feedbacks.
About the age of the population, considering also the comments of other reviewers, we decided to focalize the attention only on adolescent. In this way the age group was homogeneous. Thank you for the comment.
-Please clarify the meaning of the sentence in line 134 that reads 'The studies were included despite the study design.' Consider rephrasing for clarity.
Reply: thank you for this comment, we modified the sentence.
-In line 154, please add a period after 'in a second step, manually.'
Reply: thank you for this comment, we added the period.
In the discussion section, please clarify whether the differences mentioned in the previous studies are descriptive or statistically significant mean differences.
Reply: thank you for the comment. We added this information in the results and discussion section.
-In line 243, the study by Temple and colleagues [69] has indirect implications. Please clarify the nature of these implications.
Reply: thank you for this comment, we modified the sentence.
-In line 246, please rewrite the sentence for clarity.
Reply: thank you for this comment, we modified the sentence.
In line 270, the authors mention several negative findings. Please provide additional explanations of these findings.
Reply: We better presented this statement in the discussion. Thank you.
In line 275, when designing interventions for youth, it is important to consider their developmental stage. This is one of the reasons why physical activity guidelines for youth are more stringent than for other populations. Please add the implications of this to the literature.
Reply: thank you for this comment. The reviewer is right, we added a sentence in this regards in the limit of the study and future studies.
In lines 281-291, this section appears somewhat arbitrary. Please revise and simplify the language.
Reply: thank you for this comment, we changed and improved the sentence.
Reviewer 4 Report
Comments and Suggestions for Authors
The article "A Possible Intervention Based on Dog-Walking to Improve The Physical Level of Adolescents and Young Adults: A Scoping Review" presents an intriguing perspective on addressing sedentary behavior, particularly in the vulnerable age groups of adolescents and young adults. This scoping review delves into the potential benefits of dog ownership and walking in enhancing physical activity levels among this demographic.
One of the most commendable aspects of this article is its comprehensive analysis of existing literature, shedding light on the positive association between physical activity and dog ownership. Despite the heterogeneity of the included studies, the narrative synthesis effectively highlights the overarching trend of increased physical activity among individuals with canine companions.
The review not only underscores the significance of addressing sedentary behavior but also proposes an innovative solution that integrates pet ownership into intervention strategies. By emphasizing the potential role of dogs as companions in promoting physical activity, the study offers a fresh perspective on public health initiatives aimed at combating the growing prevalence of obesity and related health issues.
Furthermore, the article provides valuable insights into the various dimensions of well-being associated with pet ownership, extending beyond physical health benefits. From psychological support to social interaction, the presence of a dog in the household emerges as a multifaceted contributor to overall health and quality of life, particularly during the critical transitional phase of adolescence and young adulthood.
The meticulous approach to data selection and synthesis, along with the integration of diverse sources, lends credibility to the findings presented. Despite the limited number of studies available, the review effectively underscores the need for further research in this area, advocating for a more nuanced understanding of the relationship between dog ownership, physical activity, and overall health outcomes.
In conclusion, "A Possible Intervention Based on Dog-Walking to Improve The Physical Level of Adolescents and Young Adults: A Scoping Review" offers a compelling argument for integrating pet ownership into public health interventions targeting sedentary behavior. By highlighting the multifaceted benefits of canine companionship, the study not only advances our understanding of the potential mechanisms underlying this association but also underscores the importance of holistic approaches to promoting health and well-being across the lifespan.
Author Response
The article "A Possible Intervention Based on Dog-Walking to Improve The Physical Level of Adolescents and Young Adults: A Scoping Review" presents an intriguing perspective on addressing sedentary behavior, particularly in the vulnerable age groups of adolescents and young adults. This scoping review delves into the potential benefits of dog ownership and walking in enhancing physical activity levels among this demographic.
One of the most commendable aspects of this article is its comprehensive analysis of existing literature, shedding light on the positive association between physical activity and dog ownership. Despite the heterogeneity of the included studies, the narrative synthesis effectively highlights the overarching trend of increased physical activity among individuals with canine companions.
The review not only underscores the significance of addressing sedentary behavior but also proposes an innovative solution that integrates pet ownership into intervention strategies. By emphasizing the potential role of dogs as companions in promoting physical activity, the study offers a fresh perspective on public health initiatives aimed at combating the growing prevalence of obesity and related health issues.
Furthermore, the article provides valuable insights into the various dimensions of well-being associated with pet ownership, extending beyond physical health benefits. From psychological support to social interaction, the presence of a dog in the household emerges as a multifaceted contributor to overall health and quality of life, particularly during the critical transitional phase of adolescence and young adulthood.
The meticulous approach to data selection and synthesis, along with the integration of diverse sources, lends credibility to the findings presented. Despite the limited number of studies available, the review effectively underscores the need for further research in this area, advocating for a more nuanced understanding of the relationship between dog ownership, physical activity, and overall health outcomes.
In conclusion, "A Possible Intervention Based on Dog-Walking to Improve The Physical Level of Adolescents and Young Adults: A Scoping Review" offers a compelling argument for integrating pet ownership into public health interventions targeting sedentary behavior. By highlighting the multifaceted benefits of canine companionship, the study not only advances our understanding of the potential mechanisms underlying this association but also underscores the importance of holistic approaches to promoting health and well-being across the lifespan.
Thank you very much for the revision and for your time in this manuscript. We really appreciated the Reviewer comments. After the revision process, we hope the Reviewer appreciate the improvements performed.
Round 2
Reviewer 1 Report
Comments and Suggestions for Authors
The quality of the manuscript has improved. However, I still feel the need to include a paragraph in the Introduction section that addresses the tools/methodologies for assessing physical activity...
Comments on the Quality of English LanguageThe quality of the English language improved...
Author Response
"The quality of the manuscript has improved. However, I still feel the need to include a paragraph in the Introduction section that addresses the tools/methodologies for assessing physical activity..."
Reply: thank you again for the comment, it is really appreciated. We added a paragraph about the methodologies for assessing physical activity directly in the introduction, please, see the yellow part added. We hope the reviewer appreciate the work done.